

# The association between negative attention biases and symptoms of depression in a community sample of adolescents

Belinda Platt[1,2], Susannah E. Murphy[3] and Jennifer Y.F. Lau[1,4]

[1] Department of Experimental Psychology, University of Oxford, Oxford, United Kingdom
[2] Department of Child and Adolescent Psychiatry, Psychosomatics and Psychotherapy, Ludwig-Maximilian-University of Munich, Munich, Germany
[3] Oxford Centre for Human Brain Activity, University of Oxford, Oxford, United Kingdom
[4] Department of Psychology, Institute of Psychiatry, Kings College London, London, United Kingdom

Corresponding author
Belinda Platt,
belinda.platt@med.uni-muenchen.de

## ABSTRACT

Adolescence is a vulnerable time for the onset of depression. Recent evidence from adult studies suggests not only that negative attention biases are correlated with symptoms of depression, but that reducing negative attention biases through training can in turn reduce symptomology. The role and plasticity of attention biases in adolescent depression, however, remains unclear. This study examines the association between symptoms of depression and attention biases, and whether such biases are modifiable, in a community sample of adolescents. We report data from 105 adolescents aged 13–17 who completed a dot-probe measure of attention bias before and after a single session of visual search-based cognitive bias modification training. This is the first study to find a significant association between negative attention biases and increased symptoms of depression in a community sample of adolescents. Contrary to expectations, we were unable to manipulate attention biases using a previously successful cognitive bias modification task. There were no significant effects of the training on positive affect and only modest effects of the training, identified in post-hoc analyses, were observed on negative affect. Our data replicate those from the adult literature, which suggest that adolescent depression is a disorder associated with negative attention biases, although we were unable to modify attention biases in our study. We identify numerous parameters of our methodology which may explain these null training effects, and which could be addressed in future cognitive bias modification studies of adolescent depression.

## INTRODUCTION

Adolescence is a time of increased vulnerability for depression. A prospective cohort study yielded one-year point prevalence estimates of episodes of major depressive disorder (MDD) that rose dramatically from around 2% in early adolescence (ages 13–15), to 15% in middle adolescence (ages 15–18) (*Hankin et al., 1998*). While for some adolescents

these symptoms subside, for others they persist and can lead to long-term psychiatric problems (*Knapp et al., 2002*; *Weissman et al., 1999*). As emphasis grows on developing early treatments, more needs to be understood about how symptoms of depression arise and abate across this developmentally-sensitive juncture.

Prominent theories of adult depression have recently considered the association between heightened attention towards negative (versus neutral) stimuli and depression (*De Raedt & Koster, 2010*; *Ellenbogen et al., 2002*; *Koster et al., 2005*; *Peckham, McHugh & Otto, 2010*; *Shane & Peterson, 2007*), with data suggesting that attention biases may play a causal role in the onset of depression (*Browning, Blackwell & Holmes, 2013*; *Dandeneau & Baldwin, 2004*; *Dandeneau & Baldwin, 2009*; *Dandeneau et al., 2007*; *MacLeod, 2012*). In the present study, we address two research questions: (a) are symptoms of depression associated with increased attention towards negative stimuli in adolescents, as they are in adults? (b) are attention biases modifiable and associated with changes in negative affect in adolescents, as they are in adults?

## Attention biases and depressive conditions

Reviews (*De Raedt & Koster, 2010*; *Peckham, McHugh & Otto, 2010*) demonstrate overwhelming support for the presence of negative attention biases in currently depressed (*Eizenman et al., 2003*; *Gotlib et al., 2004a*; *Gotlib et al., 2004b*; *Gupta & Kar, 2012*; *Joormann & Gotlib, 2007*; *Leyman et al., 2007*; *Rinck & Becker, 2005*; *Suslow, Junghanns & Arolt, 2001*) and dysphoric (*Bradley et al., 1998*; *Ellenbogen et al., 2002*; *Koster et al., 2005*; *Shane & Peterson, 2007*) adults. Most studies have measured attention biases using the dot-probe task (*MacLeod, Mathews & Tata, 1986*), where participants are briefly exposed to a negative and a neutral stimulus presented simultaneously. A probe subsequently appears in the location of either the negative stimulus (congruent trial) or neutral stimulus (incongruent trial) and participants' reaction times (RT) to identify a characteristic of the probe (e.g., orientation) are measured. Negative attention biases are characterized by faster RTs to congruent trials and slower RTs to incongruent trials. Although some studies have failed to find evidence of negative attention biases in adults with depression (*Karparova, Kersting & Suslow, 2005*; *MacLeod, Mathews & Tata, 1986*; *Mogg et al., 1993*), this may be due to the conditions under which attention biases have been measured (*De Raedt & Koster, 2010*; *Peckham, McHugh & Otto, 2010*), for example, stimuli may need to be self-relevant. There is also ongoing debate about whether stimuli need to be exposed for more than 1,000 milliseconds (ms) in order to observe depression-related biases (*Peckham, McHugh & Otto, 2010*).

Some studies have investigated whether negative attention biases also characterize adolescents with depression. An early study using the dot-probe task to compare clinically depressed adolescents and healthy controls found no group differences in attention bias, although it should be noted that the sample was relatively small ($N = 19$ depressed participants) (*Neshat-Doost et al., 2000*). More recent dot-probe studies of larger samples suggest that depressed adolescents show attention biases towards sad (*Hankin et al., 2010*) and angry (*Salum et al., 2013*) (versus neutral) faces. Using an alternative task which measures

attentional control, the Go/No-Go task, other studies have shown speeded switching of attention to negative words (*Maalouf & Brent, 2012*) and faces (*Ladouceur et al., 2006*) in depressed versus non-depressed adolescents. It is equally important to examine the association between symptoms of depression and attention biases in non-clinical samples given overwhelming evidence that adolescent depression is a continuous (rather than discrete) disorder (*Wesselhoeft et al., 2013*), with subthreshold depression in adolescence predicting depressive disorders and suicidal behavior in adulthood (*Fergusson et al., 2005*). Few studies have explored attention biases in dysphoric adolescents (*De Voogd et al., 2014*; *Lonigan & Vasey, 2009*; *Reid, Salmon & Lovibond, 2006*), with one finding no correlation between attention bias and symptoms of depression (*De Voogd et al., 2014*), and another finding a correlation that was better explained by anxiety symptoms (*Reid, Salmon & Lovibond, 2006*). A final study did not assess the relationship with symptoms of depression, but did observe a correlation between attention bias and negative affect (*Lonigan & Vasey, 2009*). Of note, none of these studies used the faces dot-probe task, as is typical in the adult literature and has been used to demonstrate attention biases in a study of clinically depressed adolescents (*Hankin et al., 2010*). Modelling our hypotheses on findings in adults i.e., finding preferential attention engagement for negative faces, our first study hypothesis was that attention biases for negative faces (as measured using the dot-probe task) would characterize dysphoric adolescents.

## Can negative attention biases be modified such that they reduce negative affect?

If the study of attention biases and adolescent symptoms of depression is to inform treatment models, a crucial question is whether these biases can be manipulated such that they initiate affect changes. Novel experimental paradigms, referred to as Cognitive Bias Modification (CBM), have been developed to manipulate cognitive patterns (including attention biases; CBM-A) through repeated training. These paradigms could be useful for planning new interventions if induced attention biases enable changes in affect or depressive symptomology. The pioneering CBM-A paradigm is a modification of the dot-probe task (*MacLeod et al., 2002*), where the frequency of incongruent trials (trials where the probe appears in the location of the neutral stimulus) is systematically increased throughout the training session. This task reduced symptomology in students with mild-moderate symptoms of depression (*Wells & Beevers, 2010*) and adults with a previous diagnosis of depression currently in remission (*Browning et al., 2012*). A CBM-A task based on a similar attention bias measure: *Posner*'s cueing task (*Posner, 1980*), has provided more mixed results, with one study suggesting effects are dependent on depression severity (*Baert et al., 2010*).

One criticism of these two tasks is that they do not expose participants to training stimuli for long enough to facilitate attentional control (*Kruijt, Putman & Van der Does, 2013*). In addition to an attention bias towards negative information, attentional control, the ability to shift attention resources from one stimulus to another, may also be impaired in adult depression (*De Raedt & Koster, 2010*; *Sanchez et al., 2015*). Poor attentional control

has been associated with increased symptoms of depression in children and adolescents (*Muris, Meesters & Rompelberg, 2007*). A CBM-A paradigm based on the visual search task (*Hansen & Hansen, 1988*) may address this limitation. In this task participants are presented with a matrix of 15 negative faces and a single positive face. Participants learn to disengage from negative stimuli and selectively attend to the positive stimulus (identifying a smiling face as fast as they can). Compared to a control training task, the paradigm is effective in reducing negative attention biases in adults with low self-esteem (*Dandeneau & Baldwin, 2004*; *Dandeneau & Baldwin, 2009*; *Dandeneau et al., 2007*), reducing stress (*Dandeneau et al., 2007*), reducing the impact of a stress manipulation (*Dandeneau & Baldwin, 2009*), and increasing self-esteem (*Dandeneau & Baldwin, 2009*; *Dandeneau et al., 2007*).

Despite the relatively promising findings of previous CBM-A studies in relation to depression, one should remain cautious about their clinical potential, not least because at least one study failed to find positive effects on vulnerability for depression (*Kruijt, Putman & Van der Does, 2013*). In this study of dysphoric adults there was no evidence that the visual search CBM-A task modified affect (*Kruijt, Putman & Van der Does, 2013*), although the relatively small sample size ($N = 40$) and the fact that the task failed to modify attention bias in the first place, limits the interpretation of these null-effects.

Adolescence is a period of protracted brain maturation and possibly higher levels of plasticity (*Cohen Kadosh, Linden & Lau, 2013*)—therefore, we might predict that modifying biases in this age range may be more effective than those data reported in previous adult studies. On the other hand, immature pre-frontal networks in adolescence (*Nelson et al., 2005*) may reduce adolescents' ability to deploy top-down inhibitory control mechanisms that are engaged in the CBM-A paradigm; therefore, we may see weaker effects on negative affect in this age group. The developmental-appropriateness of CBM-A paradigms has been explored in relation to anxiety disorders. One review suggests that CBM-A paradigms may be effective in reducing *anxiety* in children and adolescents (*Lowther & Newman, 2014*), although a more recent review highlights the need to remain cautious about the clinical efficacy of CBM-A for youth before more robust effects on symptomology are seen (*Cristea et al., 2015*). To date, just one study has investigated CBM-A in relation to adolescent depression. De Voogd and colleagues (*2014*) administered two sessions of a visual search CBM-A training or a placebo-control training task to 32 adolescents aged 13–16. Attention biases were measured before and after training using an assessment version of the visual search training task described above, in which 50% of trials involved finding a positive face in a matrix of negative faces, and 50% of trials involved finding a negative face in a matrix of positive faces. Attention bias was calculated by subtracting mean RT to negative targets, from mean RT to positive targets. Although the CBM-A training paradigm appeared to be effective in modifying attention biases, it remains unclear whether training effects transfer to other attention bias measures e.g. the dot-probe task. Secondly, although no effect of CBM-A on symptoms of depression was observed, negative affect may have been a better outcome for detecting more subtle effects of CBM-A on depression. Our second aim was therefore to examine the efficacy of the visual search task in dysphoric adolescents, but using the faces dot-probe task to measure

the effects on attention bias. Unlike De Voogd and colleagues, we also investigated whether the training had an effect on positive and negative affect.

## Current aims and hypotheses

In a single-session CBM-A training study, we addressed two outstanding questions in the adolescent literature. First, we investigated whether negative attention biases (as measured by the faces dot-probe task) were associated with symptoms of depression in a sample of adolescents. Of note, we used a dot-probe stimulus duration of 500 ms (a) in order to facilitate comparison of the results with a previous study using the same paradigm and CBM-A procedure (*Dandeneau et al., 2007*) and (b) because we were concerned that a longer presentation time could not distinguish between attention and elaborative biases. Second, we examined whether modifying attention biases by increasing attentional control could alter negative affect in this sample. We selected the visual search CBM-A task because of studies supporting its efficacy in modifying attention biases in adults (*Dandeneau & Baldwin, 2004*; *Dandeneau & Baldwin, 2009*; *Dandeneau et al., 2007*) and adolescents (*De Voogd et al., 2014*) and because the task may enhance attentional control in participants with symptoms of depression (*Peckham, McHugh & Otto, 2010*). Again, to facilitate comparison with previous studies (*Dandeneau et al., 2007*; *De Voogd et al., 2014*), we used negative and positive (rather than negative and neutral) training stimuli in the training task. We also explored whether the effects of the attention training task were more pronounced in participants with more symptoms of depression, who were more likely to show an initial negative attention bias.

## MATERIALS & METHODS

### Participants and procedure

A community sample of 112 adolescents aged 13–17 (mean age = 16 years, 5 months; standard deviation (sd) = 0.76) were recruited through urban and rural secondary state schools and public advertisements in Oxfordshire, UK. The study session lasted 45 min. Adolescents recruited from schools were tested in groups at individual computers in a research laboratory at the University of Oxford. Adolescents recruited through public advertisements were tested individually in the same room and were given a £10 gift voucher for their participation. Participants first completed computerized measures of symptoms of depression, attention bias and affect. Participants were then randomly allocated to receive one of two CBM-A manipulations: learning to ignore emotionally *negative* (experimental group) or *neutral* (control group) stimuli. We oversampled participants in the experimental group ($N = 75$) compared to the control group ($N = 30$) because of a priori expectations that there would be more variability in responsiveness to the experimental manipulation. Attention bias and affect ratings were measured again after training. Ethical approval for the study was provided by Oxford University Central University Research Ethics Committee (MSD/IDREC/C1/2010/56) and the study was therefore performed in accordance with the ethical standards laid down in the 1964 Declaration of Helsinki and its later amendments. Participants aged 16–17 provided

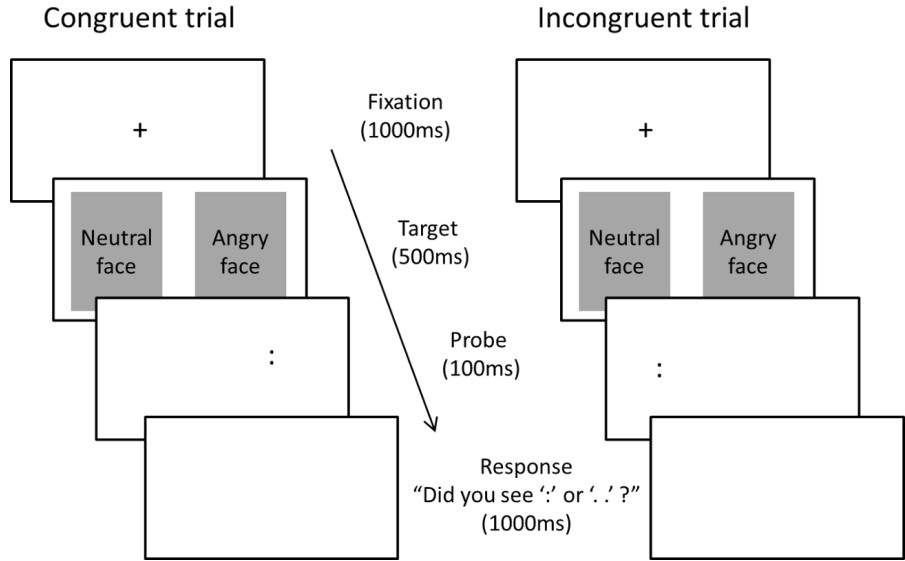

**Figure 1 Dot-probe task parameters.** The original stimuli, not displayed here due to copyright reasons, were grey-scale faces taken from the NimStim dataset (*Tottenham et al., 2009*).

written informed consent and participants aged 13–15 provided written assent (written informed consent was provided by their parents).

## Measures

### Symptoms of depression

Symptoms of depression were assessed in adolescents using the Children's Depression Inventory (CDI) (*Kovacs, 1992*) and were available for 93% ($N = 98$) of the final sample. In community samples the CDI correlates highly with other self-report measures of low mood (*Doerfler et al., 1988*). Although the original version contains 27 items, item 9 which assesses suicidal ideation was not administered here because of ethical concerns about suggesting suicide to those who might not have thought about it (*Hanley & Gibb, 2011*; *Nolen-Hoeksema, Girgus & Seligman, 1992*; *Smucker et al., 1986*). In the current sample, CDI scores ranged from 1 to 34 out of a possible 52 (mean = 12.10, $sd = 6.6$). Depression scores demonstrated high internal consistency (Cronbach's $\alpha = 0.85$). Since depression scores were non-normally distributed ($K$-$S$ test statistic = 0.14, $df = 98$, $p < 0.001$) a $\log_{10}(x + 1)$ transformation was applied and transformed scores are reported hereafter.

### Attention bias

Attention bias was assessed using a modified version of the dot-probe task (*Bradley et al., 1998*) (see Fig. 1). Stimuli were five male and five female adult grey-scale faces (participants were shown stimuli of their own gender) taken from the NimStim dataset (*Tottenham et al., 2009*). A negative (angry) and neutral expression of each face was presented side-by-side. We chose to use angry (threatening) faces for the negative stimuli in line with much of the previous literature and because threatening faces are thought to elicit a depressive reaction by depicting personal rejection whereas sad faces may depict the emotion of an

observer (*Leyman et al., 2007*). Whether the negative face appeared on the left or the right hand side was counterbalanced across trials. The face pairs were of resolution 505 dpi, measured 200 x 257 pixels, and were displayed on a black background. The face stimuli were presented for 500 ms and were immediately followed by a probe, which appeared in the location of one of the preceding faces. The probe was two dots presented either vertically (':') or horizontally ('..') and was displayed on the screen for 100 ms. Participants were required to identify the probe orientation using the 'z' key for vertical orientation (':') and the 'm' key for horizontal orientation ('..'). The probe was presented for 100 ms, rather than an unlimited or longer time interval, to prevent elaborative stimulus processing and support the automatic attention process we were trying to capture. Responses made within 1,000 ms of probe onset were recorded. The inter-trial interval varied randomly between 500 ms and 1,250 ms. Face-pairs and probes were presented in a random order. There were two trial-types: congruent (the probe replaced the negative face) and incongruent (the probe replaced the neutral face). In total there were 80 test trials and 40 'filler' trials (where neutral expressions of each face were presented side by side). Trials were presented in two blocks of 60 trials, preceded by 12 practice trials which gave participants 'correct' or 'incorrect' feedback. The tasks were programmed using the E-Prime 2.0 software (*Psychology Software Tools, Inc., 2012*).

### Affect ratings

Affect ratings were assessed using the Positive and Negative Affectivity Scale (PANAS) (*Watson, Clark & Tellegen, 1988*). The 20-item scale provides both a positive and negative affect score. The scale has high reliability (Cronbach's $\alpha = 0.83$) in adolescent samples (*Garcia et al., 2012*). Affect ratings for both time points were available for 98% ($N = 103$) of the final sample.

## Experimental and control training tasks

Both training tasks involved identifying a distinguishing feature in a $4 \times 4$ grid, over six practice trials and a single block of 112 experimental trials (*Dandeneau et al., 2007*) (see Fig. 2). In both conditions, trials began with a fixation cross ('+') which appeared in the center of the screen for 1,000 ms. This was followed by the $4 \times 4$ grid (10,000 ms). Participants were asked to identify the target stimuli using the left button of the mouse. For the experimental training task, the target stimulus was a positive (smiling) face, while the distracting stimuli were negative (frowning) faces. For the control training task, the target stimulus was a five-petalled flower while the distracting stimuli were seven-petalled flowers. Practice trials provided participants with 'correct' or 'incorrect' feedback.

Stimuli in the experimental training condition were modified from the original task to include 16 adolescent (instead of adult) faces selected from the NIMH Child Emotional Faces Picture Set (NIMH-ChEFS) (*Egger et al., 2011*). The faces were of resolution 300 dpi, measured $8.5 \times 8.5$ mm on the screen, and were presented in color on a grey background. Each of the smiling faces was presented seven times in each of the 16 positions (112 trials). Pictures of the five- and seven-petalled flowers were of the same resolution and size as the

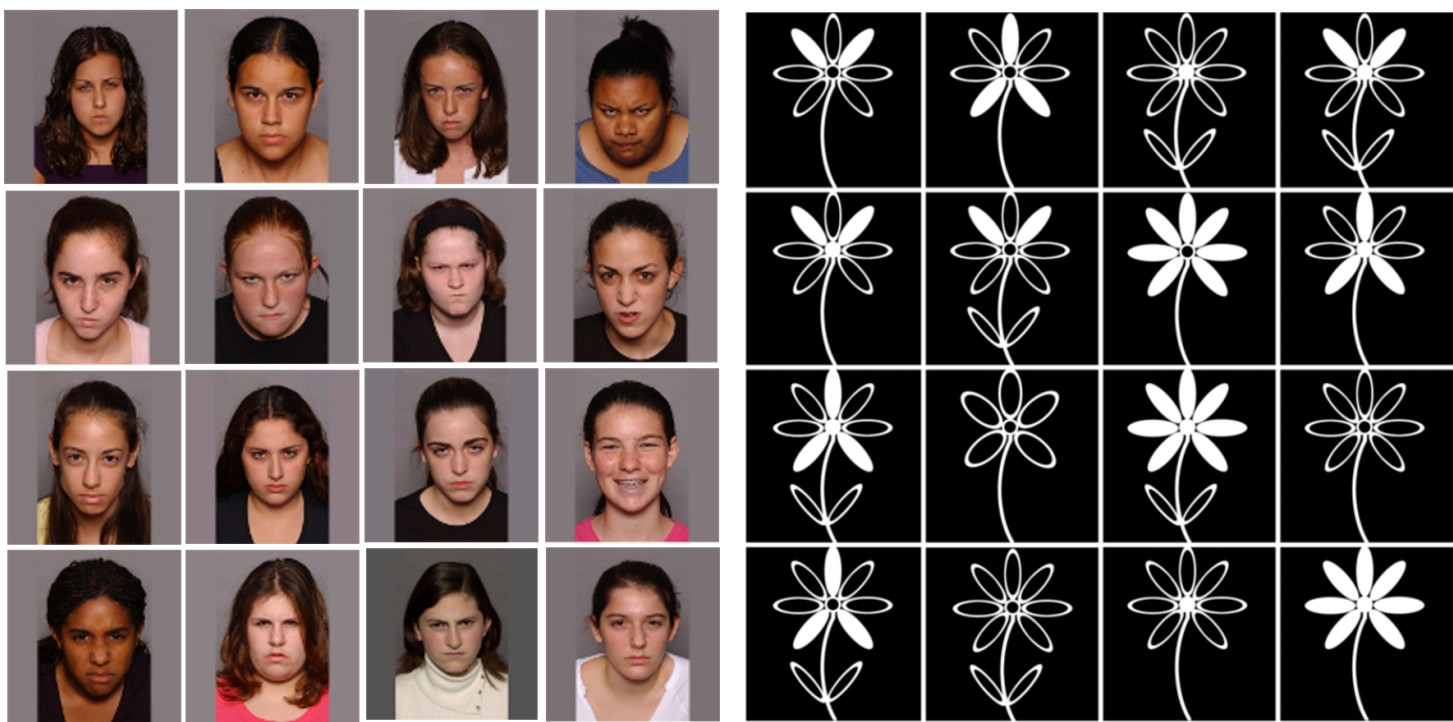

**Figure 2** Experimental and control cognitive bias modification of attention (CBM-A) training tasks.

adolescent face stimuli and presented for the same duration. The tasks were programmed using the E-Prime 2.0 software (*Psychology Software Tools, Inc., 2012*).

## Data preparation and statistical analysis

Attention bias was calculated by subtracting mean RT (ms) to congruent trials from incongruent trials, such that positive scores represented a negative attention bias (*Dandeneau et al., 2007*; *MacLeod, Mathews & Tata, 1986*). Trials were excluded from analysis if the response was inaccurate or if RT was less than 200 ms or greater than two standard deviations from each participant's mean RT (*Ratcliff, 1993*; *Roy et al., 2008*). Of note, the mean error rate for dot-probe trials was 16.60% ($sd = 9.1$). Dot-probe data from five participants (either pre- or post-training task data) were incomplete due to a technical error during runtime and these subjects' complete data were therefore removed. There was no reason to believe there was a systematic pattern to the disruption and therefore the exclusion of these data is unlikely to have caused a bias in the final sample. Two participants in the control training condition were excluded because they obtained less than 65% accuracy on the task.

Data were analyzed using SPSS. In order to assess whether participants with more symptoms of depression demonstrated a greater negative attention bias, a bivariate correlation analysis between (transformed) symptoms of depression and baseline attention bias was conducted. To test the main effect of the training manipulation (experimental, control) on attention bias change scores (post-training-bias minus baseline-bias score) and the interaction between training manipulation and (transformed) symptoms of

**Table 1 Participant characteristics.**

| | Whole sample (105) mean (sd) | Experimental condition (N = 75) mean (sd) | Control condition (N = 30) mean (sd) |
|---|---|---|---|
| Age (N = 104) | 16.39 (0.8) | 16.43 (0.8) | 16.31 (0.8) |
| Number (and percentage) of females (N = 104) | 92 (87.6) | 67 (89.3) | 25 (83.3) |
| Depressive symptoms (N = 98) | 12.10 (6.6) | 11.26 (5.8) | 14.00 (7.8) |
| Baseline attention bias (ms) (N = 105) | 0.31 (26.5) | −1.24 (26.4) | 4.20 (26.8) |
| Post-training attention bias (ms) (N = 105) | 1.57 (23.0) | 1.14 (22.1) | 2.66 (25.5) |
| Baseline positive affect (N = 104) | 5.59 (1.6) | 5.60 (1.7) | 5.56 (1.5) |
| Post-training positive affect (N = 104) | 5.68 (1.7) | 5.71 (1.7) | 5.60 (1.7) |
| Baseline negative affect (N = 104) | 1.93 (1.5) | 1.77 (1.4) | 2.33 (1.7) |
| Post-training negative affect (N = 104) | 1.71 (1.5) | 1.46 (1.3) | 2.34 (1.9) |
| CBM-A trial accuracy (%) (N = 105)[a] | 96.2 (4.6) | 97.2 (4.2) | 93.6 (4.7) |
| CBM-A trial RT (N = 105)[a] | 2,848.1 (615.9) | 2,721.4 (590.2) | 3,165.0 (571.2) |

Notes.

[a] Significant difference between those in the experimental (N = 75) and Control (N = 30) conditions ($p < 0.05$).

depression (continuous variable), a repeated-measures custom model ANOVA was used. Two similar custom model ANOVA tests explored the effects of training manipulation and symptoms of depression on change scores for negative affect and positive affect respectively. Differences in baseline demographic characteristics and training performance between the experimental and control groups were assessed using chi-square and $t$-tests.

## RESULTS

The final sample comprised 105 adolescents (Table 1).

### What is the relationship between symptoms of depression and attention bias?

A bivariate correlation (performed on data from the 98 (93%) participants who provided depressive symptom scores) revealed that symptoms of depression correlated significantly with baseline attention bias (Pearson's $R = 0.20$, $p = 0.05$; Fig. 3). Of note, symptoms of depression were also associated with negative ($N = 97$; Pearson's $R = 0.41$, $p < 0.001$) and positive ($N = 97$; Pearson's $R = -0.46$, $p < 0.001$) affect. However, there was no evidence of a significant association between baseline attention bias and negative affect ($N = 104$; Pearson's $R = 0.09$, $p = 0.36$), or between baseline attention bias and positive affect ($N = 104$; Pearson's $R = 0.17$, $p = 0.09$).

### Can attention biases be modified such that they improve negative affect?

There were no significant differences between the two training conditions in terms of participants' age ($t_{102} = 0.70$, $p = 0.48$), symptoms of depression ($t_{96} = -1.53$, $p = 0.13$), baseline attention bias ($t_{103} = -0.95$, $p = 0.34$), positive affect ($t_{102} = 0.10$, $p = 0.92$), or negative affect ($t_{45.45} = -1.56$, $p = 0.13$; Table 1). There was no evidence of gender differences (data available for $N = 104$) between the experimental training condition
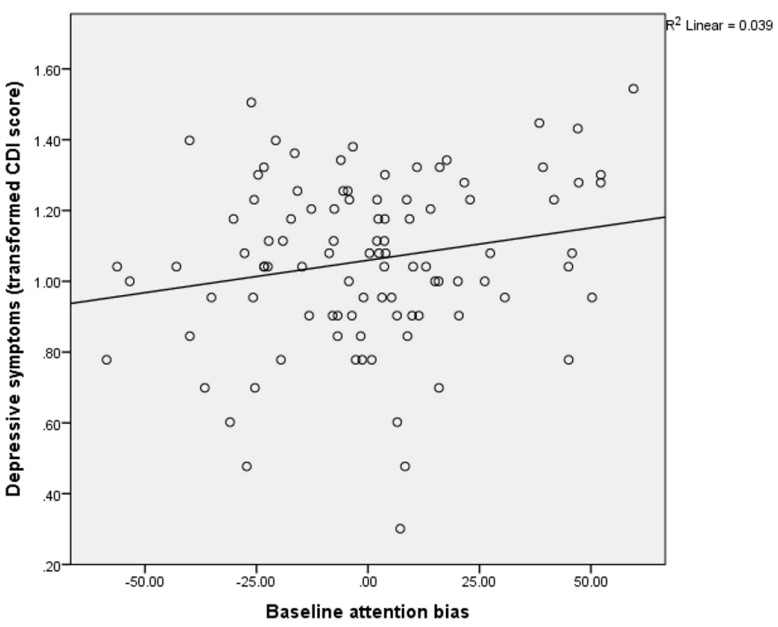

**Figure 3 Association between baseline attention bias and depressive symptoms.**

and the control condition ($\chi^2 = 1.09$, $p = 0.30$; Table 1). Task performance varied significantly between those who completed the experimental and those who completed the control training. Participants who performed the control training task were less accurate ($t_{48.4} = 3.70$, $p < 0.001$, Cohen's $d = 0.8$) and took longer to correctly identify the target stimulus ($t_{103} = -3.51$, $p < 0.001$, Cohen's $d = 0.8$) than those who performed the experimental training task.

### Change in attention bias

The ANOVA model (data on all relevant variables available for $N = 98$ participants) revealed no main effect of condition on change in attention bias ($F_{1,94} = 0.04$, $p = 0.85$), suggesting that attention biases did not change as a function of whether individuals received experimental or control training. Neither was there a main effect of symptoms of depression ($F_{1,94} = 1.65$, $p = 0.20$) or a significant two-way interaction between condition and symptoms of depression ($F_{1,94} = 0.01$, $p = 0.93$).

### Change in affect

An ANOVA model (data on all relevant variables available for $N = 96$ participants) revealed no main effect of condition on change in negative affect ($F_{1,92} = 2.56$, $p = 0.11$). There was also no evidence of a main effect of symptoms of depression ($F_{1,92} = 0.29$, $p = 0.59$) on change in negative affect. A two-way interaction between condition and depressive symptoms was marginally significant ($F_{1,92} = 3.93$, $p = 0.05$, partial eta squared $= 0.04$), suggesting that the effect of the training task on change in negative affect varied as a function of participants' symptoms of depression. In order to explore this interaction, a median-split variable was created based on (transformed) depressive symptom scores (median $= 1.06$)—and the effect of training condition on change in

negative affect was investigated using paired-samples $t$-tests in each subsample separately. For participants with more symptoms of depression, those in the experimental condition showed a significant reduction in negative affect following the training (mean $= -0.44$, $sd = 1.1$, $t_{30} = 2.24$, $p = 0.03$, Cohen's $d = 0.3$), whereas depressed participants in the control condition showed no significant change ($t_{17} = -1.30$, $p = 0.21$). Participants with fewer symptoms of depression showed no significant change in negative affect in the experimental condition ($t_{34} = 1.40$, $p = 0.17$) or control condition ($t_{11} = 1.86$, $p = 0.09$).

A similar ANOVA (data on all relevant variables available for $N = 96$ participants) revealed no main effect on change in positive affect of condition ($F_{1,92} = 0.26$, $p = 0.62$), no main effect of symptoms of depression on change in positive affect ($F_{1,92} = 0.04$, $p = 0.85$), and no interaction between condition and symptoms of depression on change in positive affect ($F_{1,92} = 0.29$, $p = 0.59$).

## DISCUSSION

Our findings demonstrate an association between symptoms of depression and negative attention bias in a community sample of adolescents. The attention bias training task used in the current study was unsuccessful in modifying attention bias or positive affect. Modest effects of the training on negative affect were observed for participants with more symptoms of depression, however these findings are interpreted with caution due to the post-hoc nature of analysis.

This study is the first to demonstrate a positive association between negative attention biases and symptoms of depression in an unselected adolescent sample. This overall finding is in line with previous studies of attention biases in clinically depressed (*Eizenman et al., 2003*; *Gotlib et al., 2004a*; *Gotlib et al., 2004b*; *Gupta & Kar, 2012*; *Joormann & Gotlib, 2007*; *Leyman et al., 2007*; *Rinck & Becker, 2005*; *Suslow, Junghanns & Arolt, 2001*) and dysphoric (*Bradley et al., 1998*; *Ellenbogen et al., 2002*; *Koster et al., 2005*; *Shane & Peterson, 2007*) adults. Furthermore, it extends a study of adolescents which found an association between attention bias and negative affect (*Lonigan & Vasey, 2009*) and supports an adolescent extension of cognitive theories, in which depression is associated with attention (albeit weakly) as well as elaborative (e.g., memory and interpretive) biases (*Beck, 1967*; *Mathews & MacLeod, 2005*; *Williams et al., 1997*).

The size of the correlation between attention bias and depressive symptoms was nevertheless relatively small (Pearson's $R = 0.2$), which may be related to the low reliability of the dot-probe task. Although the dot-probe has been established as the standard measure of attention bias, developing new measures which more reliably measure attention bias in youth depression is important for future research in this field. Studies of attention bias in adult depression have recently used eye-tracking during passive viewing tasks. In a recent study of alcohol-related attention biases, eye-tracking was found to have superior internal reliability over traditional behavioural reaction-time measures (*Christiansen et al., 2015*). A meta-analysis of eye-tracking studies suggests the depressed versus non-depressed adults show increased maintenance of attention towards dysphoric stimuli during passive-viewing (extended presentation of multiple emotional stimuli

without instruction) (*Armstrong & Olatunji, 2012*). The only study to date of youth depression found the opposite effects—that youth with depression showed attentional avoidance (reduced maintenance) with sad stimuli (*Harrison & Gibb, 2014*). These findings indicate that eye-tracking may be a valuable tool for investigating attention biases in youth depression, and that there is more to be understood about the developmental aspects of attention biases in depression.

Future studies of restricted periods of childhood and adolescence could also help identify developmental periods when associations between attention biases and symptoms of depression first emerge. Our data suggest that by adolescence hypervigilance of negative cues is already characteristic of depressive symptoms, but it is not clear *when* this linkage first emerges. It should be noted here that in our version of the dot-probe task the probe was displayed for a limited duration of 100 ms. This duration is shorter than in many previous dot-probe studies and may have yielded a somewhat elevated error rate (mean error rate = 16.60%, $sd$ = 9.1). However, we also cannot exclude the possibility that the higher than expected error rate was due to our sample being relatively young. Of note, a recent study by Britton and colleagues (*2013*), also delivering the dot-probe task to typically developing adolescents, found mean error rates (before the exclusion of outliers) of 16–18%.

A critical clinical question is whether modifying attention biases has therapeutic potential for adolescents. CBM-A approaches were initially a topic of great excitement, with early studies demonstrating significant effects of training on negative mood and vulnerability for adult depression (*Baert et al., 2010*; *Browning et al., 2012*; *Dandeneau & Baldwin, 2004*; *Dandeneau & Baldwin, 2009*; *Dandeneau et al., 2007*; *MacLeod, 2012*; *Wells & Beevers, 2010*). CBM-A paradigms have potential as an add-on to standard psychiatric treatment given the ease by which they can be administered and the fact that they complement existing treatment approaches by addressing more automatic information processing. Indeed, one of the first meta-analyses of CBM-A for anxiety suggested medium ($d$ = 0.61) effect sizes, which may be even larger in clinical populations (*Hakamata et al., 2010*) and are comparable with effect sizes of existing treatments for anxiety. More recent meta-analyses of CBM-A for anxiety and depression in adult (*Cristea, Kok & Cuijpers, 2015*; *Heeren et al., 2015*) and youth (*Cristea et al., 2015*; *Lowther & Newman, 2014*) populations rightly call for caution in the recommendation of CBM-A paradigms as clinical tools, since effects are not replicated in all studies, effect sizes are sometimes small, and studies which trial the effects of CBM-A paradigms are often poorly conducted or provide insufficient data to assess the methodological quality. Thus, the lack of positive effects of CBM-A on emotional vulnerability is not reason enough for rejecting the potential of CBM-A paradigms. Clarke and colleagues (*2014*) observed that of the 29 CBM-A studies which reported data on attention bias change (in addition to effects on emotional vulnerability), 26 showed that when attention bias was successfully changed, emotional vulnerability was modified too ($N$ = 16), and when attention bias was unsuccessfully modified, there was no effect on emotional vulnerability ($N$ = 10). Together these findings suggest that CBM-A paradigms do indeed have potential as clinical tools,

but that more research is needed to understand why they are often ineffective in modifying attention biases in the first place.

Just one study has measured the effects of a CBM-A task on vulnerability for depression in youth (*De Voogd et al., 2014*). Although this study is one of few to successfully modify attention bias without seeing an effect on emotional vulnerability, emotional vulnerability was measured as depressive symptoms rather than positive or negative affect (which may be more sensitive measures for detecting change). The CBM-A task was also designed to modify social anxiety rather than depression.

In our study we chose to use the visual search task because of its previous success in modifying attention biases and affect in adults and adolescents (*Dandeneau & Baldwin, 2004*; *Dandeneau & Baldwin, 2009*; *Dandeneau et al., 2007*; *De Voogd et al., 2014*), and because we hypothesized that the training could enhance general attentional control, which may in turn affect attentional engagement with negative stimuli specifically. However, we failed to find evidence that negative attention biases could be manipulated using a visual search attentional bias training task in our young sample. This is the second published failure to replicate the effects of this training task on attention bias (*Kruijt, Putman & Van der Does, 2013*). Nevertheless, before concluding that attention biases cannot be modified using this visual search paradigm in dysphoric adolescents, numerous alternative explanations are worth considering. One possibility is that attentional training effects simply do not transfer from one task to another i.e., whilst the dot-probe task measures attentional *engagement with negative* stimuli, the visual search CBM-A task trains *engagement with positive* stimuli and *disengagement from negative* stimuli. This may explain why De Voogd and colleagues (*2014*) found effects of the visual search CBM-A task on attention bias but we and others (*Kruijt, Putman & Van der Does, 2013*) have not. Indeed, although we chose to use the dot-probe task to aid comparison with the previous literature, it has been shown to have relatively low reliability, particularly in non-clinical samples (*Schmukle, 2005*). A second possible explanation is that we trained attention to (coloured) adolescent faces, but measured attention bias to (grey-scale) adult faces. We may therefore have trained an attention bias towards salient (coloured, age-adapted) stimuli specifically, which did not transfer to less salient (grey-scale, adult) stimuli. The failure to include a visual search measure of attention bias, and the lack of consistency in test stimuli, are therefore limitations of the current study. A third possible explanation for our findings is that training effects may only be expected in participants showing an attention bias at baseline. Using the same ANOVA method reported for the full sample above, we explored post-hoc whether participants with baseline attention bias (i.e., dot-probe bias score > 0; $N = 53$) benefited from training. However, there remained no evidence of an effect of training group on any of the three outcome measures (change in attention bias, negative mood, or positive mood), or of an interaction between training group and symptoms of depression (all $Ps > 0.2$). A fourth possible explanation for the lack of effects is that multiple training sessions may be needed to elicit robust effects on attention bias. The single-session nature of the CBM-A paradigm is another limitation of the current study.

Some evidence to support the beneficial effect of the CBM-A task is supported by the fact that the experimental training condition was associated with reduced negative affect (in participants with higher levels of depressive symptoms). However, whilst it is possible that this effect is mediated by a change in attention bias that was not detected by the dot-probe assessment task, changes in affect may also have been due to non-specific differences (e.g., task difficulty) between the two training conditions. Indeed, RT and accuracy data suggest that the experimental (faces) task was significantly easier to perform than the control (flowers) task, replicating findings from a recent study using the same CBM-A training tasks (*Kruijt, Putman & Van der Does, 2013*). Furthermore, when we added training task accuracy scores to our ANOVA model, thereby assessing whether it modified the effect of training on negative affect, the previously significant interaction between condition and symptoms of depression disappeared. Although it seems plausible that participants with more symptoms of depression were simply more rewarded from successfully completing the experimental task, since the condition × depression × training accuracy interaction was non-significant, we remain cautious in this interpretation due to the post-hoc nature of the analyses. Finally, it should be acknowledged that the sample sizes in the post-hoc *t*-test analyses were unequal due to oversampling in the experimental group. Smaller sample sizes in high depressed ($N = 18$) and low depressed ($N = 12$) participants allocated to the control condition could explain the lack of training effects in these groups.

Our study highlights the fragility of CBM-A data and the infancy of the attention bias literature in relation to adolescent depression. However, whilst remaining cautious about the clinical potential of CBM-A tasks for adolescent depression, the positive findings from CBM-A studies of adult depression, along with the finding that CBM-A paradigms generally modify emotional vulnerability when attention bias is successfully modified, suggests that exploration of the optimal paradigms and parameters needed for attention bias modification in adolescents is a worthy area of future research (*Clarke, Notebaert & MacLeod, 2014*). There are numerous ways in which future CBM-A studies could be conducted in order to increase the chances of attention bias change. Firstly, CBM-A tasks may be more successful in altering biases if multiple training sessions are employed. Secondly, given our finding that symptoms of depression are associated with baseline attention bias, CBM-A studies of adolescents with more severe symptoms of depression may show stronger training effects. On the other hand, depression-related attentional control difficulties could mean that these participants show more difficulty in performing the CBM-A task. CBM-A studies of clinically depressed adolescents would nevertheless be of interest. Thirdly, Posner's cueing task has shown promise in the modification of attention biases in depressed adults, albeit with positive effects on symptom severity emerging following a post-hoc analysis (*Baert et al., 2010*), as in the current study.

## CONCLUSION

Adolescence is a period of vulnerability for depression, yet little is known about the role of negative information processing in the onset of depressive symptoms during this

developmental period. Data from an unselected sample of adolescents suggest that, as has been demonstrated in adult studies, negative attention biases are associated with increased symptoms of depression. In contrast to previous studies, we found no evidence that the visual-search CBM-A task could modify attention biases (as measured using the dot-probe task) or positive affect in our community sample of adolescents. Modest effects of the training on negative affect were observed for participants with more symptoms of depression, however these findings are interpreted with caution due to the post-hoc nature of analysis. Numerous complexities associated with measuring and modifying attention biases mean that further examination of these effects is needed before firm conclusions about the precise role of attention biases in adolescent depression, and their implications for clinical practice can be drawn.

## ACKNOWLEDGEMENT

We thank the participants who gave their time to take part in this study and their parents and teachers for facilitating their involvement. We are also grateful to Kevin Hilbert, Merel Kerstholt, Marissa Waldemore and Sophie Raeder for their help in data collection.

### Funding

This research is funded by a Young Investigator Award from the Brain and Behaviour Research Foundation, and research funding from the Calleva Research Centre, both awarded to JYFL. SM is supported by the National Institute for Health Research (NIHR) Oxford Biomedical Research Centre based at Oxford University Hospitals Trust Oxford University. The funders had no role in study design, data collection and analysis, decision to publish, or preparation of the manuscript.

### Grant Disclosures

The following grant information was disclosed by the authors:
Brain and Behaviour Research Foundation.
Calleva Research Centre.
National Institute for Health Research (NIHR) Oxford Biomedical Research Centre.

### Competing Interests

SM has served as a consultant to p1Vital and has participated in paid speaking engagements for Eli Lilly and Co. UK. BP, SS and JYFL declare that they have no conflict of interest.

### Author Contributions

- Belinda Platt conceived and designed the experiments, performed the experiments, analyzed the data, contributed reagents/materials/analysis tools, wrote the paper, prepared figures and/or tables, reviewed drafts of the paper.

- Susannah E. Murphy conceived and designed the experiments, analyzed the data, contributed reagents/materials/analysis tools, wrote the paper, reviewed drafts of the paper.
- Jennifer Y.F. Lau conceived and designed the experiments, performed the experiments, analyzed the data, contributed reagents/materials/analysis tools, wrote the paper, reviewed drafts of the paper.

## Human Ethics

The following information was supplied relating to ethical approvals (i.e., approving body and any reference numbers):

Oxford University Central University Research Ethics Committee MSD/IDREC/C1/2010/56.

## Supplemental Information

Supplemental information for this article can be found online at http://dx.doi.org/10.7717/peerj.1372#supplemental-information.

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
