# Peer review of "The association between negative attention biases and symptoms of depression in a community sample of adolescents"

_PeerJ, doi:10.7717/peerj.1372_

## Round 0.1 · original submission · Major Revisions

· Academic Editor

Major Revisions

Please address all comments raised by the reviewers and provide a point-by-point response indicating how and where each issue was addressed in the revised manuscript. Please pay particular attention to the revisions requested for the introduction (to better cover the current literature on the topic) as well as the Methods, Results and interpretations in the Discussion section.

Reviewer 1 ·

Basic reporting

I enjoyed reading this well-written paper on attentional biases and depressive symptoms in adolescents. The report of this non-replication could make a valuable contribution to the literature. I have only a few minor remarks.
Lines 57-58, 62-65: Please consider here studies reporting findings that do not suggest that attention biases play a causal role in depression or are associated with changes in negative affect in adults (as included later in the article). Provide a balanced overview of the literature.
Line 67: see another meta-analysis on this topic (Armstrong & Olatunji, 2012). Please consider insights from research using eye tracking.
Line 130-131: see Sanchez et al. for a recent study examining ability related processes in attentional allocation (Sanchez, Everaert, De Putter, Mueller, & Koster, 2015).

References
Armstrong, T., & Olatunji, B. O. (2012). Eye tracking of attention in the affective disorders: A meta-analytic review and synthesis. Clinical Psychology Review, 32(8), 704–723. doi:10.1016/j.cpr.2012.09.004
Sanchez, A., Everaert, J., De Putter, L. M. S., Mueller, S. C., & Koster, E. H. W. (2015). Life is…great! Emotional attention during instructed and uninstructed ambiguity resolution in relation to depressive symptoms. Biological Psychology, 109, 67-72. doi:10.1016/j.biopsycho.2015.04.007.

Experimental design

no comments

Validity of the findings

no comments

·

Basic reporting

The manuscript is well-written in a clear style. The literature should be updated, to more appropriately reflect recent meta-analyses carried out in the field of attentional bias modification in children and adolescents. This would also give the 'null findings' a better base, since the effects of attentional bias modification in the literature are generally small, and this paper is no exception.

Experimental design

Quite some details on the experimental setting are missing, while they are important in reaction-time based studies. Was the experiment conducted in a lab? Where the participants payed to enter the study? Details about the demographics of the participants are lacking.

The experimental design does seem appropriate to test the current hypotheses, although some decisions were made that made the design suboptimal, such as testing attentional bias with grayscale adult pictures, and training with adolescents. These limitations are however noted by the authors.

Validity of the findings

The sample is appropriately large to support the conclusions.

However, my main problem with this manuscript is that the hypotheses aren't confirmed, but discussion and abstract do seem to tell otherwise. It would be better for the field to honestly publish the findings, that the relationship between depression symptoms and attentional bias is weak (this might be just a spurious finding based on change, with an unreliable instrument), that the experiment didn't alter attentional bias (at least not detectable by the dot probe), and that only in a subsample effects were found on depression, but not on the more sensitive positive and negative affect scales.

Which, altogether and in the light of the recent meta-analysis, disencourages the use of the dot-probe and does not support clinical application of CBM-A in adolescents.

Additional comments

I do think this is well-conducted research, but I think the results should be presented more balanced, reflecting the null-findings. These kinds of studies do add to the puzzle of attentional bias and may help others in their study design, therefore this manuscript should get a place somewhere.

---

## Round 0.2 · accepted · Accept

· Academic Editor

Accept

The issues raised by reviewers have now been adequately addressed.

Reviewer 1 ·

Basic reporting

No Comments

Experimental design

No Comments

Validity of the findings

No Comments

Additional comments

The authors have adequately addressed my prior comments.

·

Basic reporting

Current developments in the literature are now accurately presented. The manuscript is well-written and provides important information for other researchers in the field. All remarks have been adequately addressed and/or explained in the rebuttal letter.

Experimental design

Information about the design has been added to this revised document. Some choices in the design may have influenced the results (such as grayscale pictures) but are clearly addressed and explained in methods and discussion, and will help other researchers in their choices when setting up similar experiments.

Validity of the findings

The findings are now presented balanced, in context.

Additional comments

Please check if the abstract in the submission system; it is different from the abstract in the manuscript.